# Students and examiners perception on virtual medical graduation exam during the COVID-19 quarantine period: A cross-sectional study

**Nazdar Ezzaddin Alkhateeb**[1]*, **Baderkhan Saeed Ahmed**[2], **Namir Ghanim Al-Tawil**[3], **Ali A. Al-Dabbagh**[1]

**1** Department of Medical Education, College of Medicine, Hawler Medical University, Erbil, Kurdistan Region, Iraq, **2** Department of Surgery, College of Medicine, Hawler Medical University, Erbil, Kurdistan Region, Iraq, **3** Department of Community Medicine, College of Medicine, Hawler Medical University, Erbil, Kurdistan Region, Iraq

* nazdar.alkhateeb@hmu.edu.krd

## Abstract

**Data Availability Statement:** All relevant data are within the paper and its Supporting Information files.

### Background

With the emergence of the COVID-19 pandemic and lockdown approach that was adopted all over the world, conducting assessments while maintaining integrity became a big challenge. This article aims at sharing the experience of conducting an online assessment with the academic community and to assess its effectiveness from both examiners' and students' perspectives.

### Methods

An online assessment was carried out for the final year medical students of Hawler Medical University/Iraq during the lockdown period of the COVID-19 pandemic, June 2020. Then, an online questionnaire was sent to a sample of 61 examiners and 108 students who have been involved in evaluating the mentioned assessment process. Mann-Whitney and Kruskal-Wallis tests were used to compare the mean ranks of the overall satisfaction scores between categories of the students and examiners. Categorical data were summarized and presented as frequencies and percentages.

### Results

The response rates among examiners and students were 69.4% and 88.5% respectively. The majority of the examiners were generally satisfied with the online examination process compared to only around a third of the students. However, both examiners and students agreed that online examination was not suitable for assessing the physical examination skills.

**Funding:** The authors received no specific funding for this work.

**Competing interests:** The authors declared that they have no competing interests.

## Conclusion

The online assessment can be considered a good alternative and acceptable method for medical students' assessment in unpredicted emergencies, yet it was not applicable in testing physical examination skills.

## Introduction

The COVID-19 pandemic has affected every part of societies all around the world, it has caused the largest interruption of education systems in the history of mankind involving almost 1.6 billion learners in over 200 countries [1]. The standard provision of medical education for undergraduate medical students has to be changed during this global crisis, and, it is important to develop innovative approaches of assessment that adopt the standards of medical education and consider the current social and environmental constraints caused by the COVID-19 pandemic [2].

Assessment is the measurement of the students' learning and the way they are assessed usually influence what and how they learn. In many countries, clinical and written examinations have been suspended, cancelled, or replaced by online examinations or new methods of assessment including; Comprehensive high-stake online exams [3], Modified national Objective Structured Clinical Examination (OSCE) [4], High stakes modified OSCE [5], Web-based OSCE [6], Online oral [7], and Online assessment [8].

College of Medicine at Hawler Medical University has considered it essential to set up a modified online examination not only to maintain the students' academic progress but also to provide enough medical graduates. Therefore, this study aims at sharing the experience of implementing a comprehensive online examination to final year students at the college of medicine during this pandemic and to assess its effectiveness and satisfaction by examiners and students.

## Materials and methods

### Study design and setting

This was a cross-sectional study performed at the College of Medicine, Hawler Medical University, Kurdistan Region/Iraq during June 2020.

### Study participants and sampling

The study population was 72 examiners and 150 6[th] year students who were involved in the online assessment experience. The sample size was estimated using the Epi info 7 computer program (free program issued by WHO and CDC) where the following information was entered into the program: the size of the population (mentioned above), the confidence interval was set at 95%, the estimated prevalence was set at 50% and the absolute precision was set at 5%. Accordingly, the estimated sample size was found to be 108 students and 61 examiners. Simple random sampling was used to select the sample out of the examiners and students' populations. The aim of the study was clarified to the participating students and examiners, and their consent was taken.

### Structure of the online assessment

Over the past years (before the COVID-19 pandemic), the final year medical students' assessment was composed of a two-part assessment; advanced clinical competence which consisted

of: Objective Structured Clinical Examination (OSCE) that assesses the skills and a written component in a form of Single best answer (SBA) and Extended matching questions (EMQ) which assess the knowledge.

It's worth mentioning that the attitude of the students is assessed as part of daily evaluation through all the courses. The college has developed a comprehensive multiform examination which include all disciplines (Pediatric, Obstetrics & Gynecology, Medicine and Surgery). This examination represents 40% of the students' final graduation average grade of the MBChB program and the remaining 60% is collected from students' grades in years 1–5.

During the COVID-19 pandemic lockdown, final assessment with the physical presence of students was not possible, therefore the curriculum and assessment committees of the college made a modification for the assessment process with taking the examiners' opinions into consideration.

## Planning and preparation

A committee (including the Dean, head of clinical departments, head of the Medical Education department, and the 6th year director) was formed to decide on the forms and number of stations, competencies to be assessed, assure that the blueprint was properly sampling the essential part of the curriculum and responsible on students' and examiners' orientation on the final exam process during COVID-19 pandemic. A grading score checklist was prepared for all sets of questions.

The 6th year director prepared 18 examination panel teams. Each team was composed of four members from major clinical specialties. A focal point was allocated for each team.

The 6th year director conducted a Zoom meeting with the 18 focal points to describe the implementation process, and discuss challenges expected and possible solutions.

To decrease the possible internet problems, the exam panel teams were asked to attend the college and to meet the students in one Zoom account (focal point's account).

## Description of the online assessment

In usual situations, clinical competencies in history taking, examination, communication, and procedural skills besides data interpretation and management competencies were assessed through OSCE stations.

In the modified online assessment, all the above-mentioned competencies were assessed except for the examination and procedural skills competencies.

The oral exam was conducted as an online Zoom meeting between the student and the examination panel. In a random manner, eighteen groups of students were created and they were distributed to the online groups (8–9 students were put into the online zoom groups). As a result, first students in each of the 18 groups would be examined with the same questions, and so on (Fig 1).

The Blueprint was regulated according to the competencies for each round of student exams, and the questions included emergency patient scenarios in different disciplines and different systems.

For the security purpose of the online exam, 10 sets of different questions according to the blueprint competencies were prepared. The 9th and 10th set were prepared as backup sets in case any connection problem appeared during the exam. In case any student experienced connection problems, they would be shifted to the end with a new set of questions i.e. 9th and 10th in order not to affect the sequence of the exam in the other examination panels.

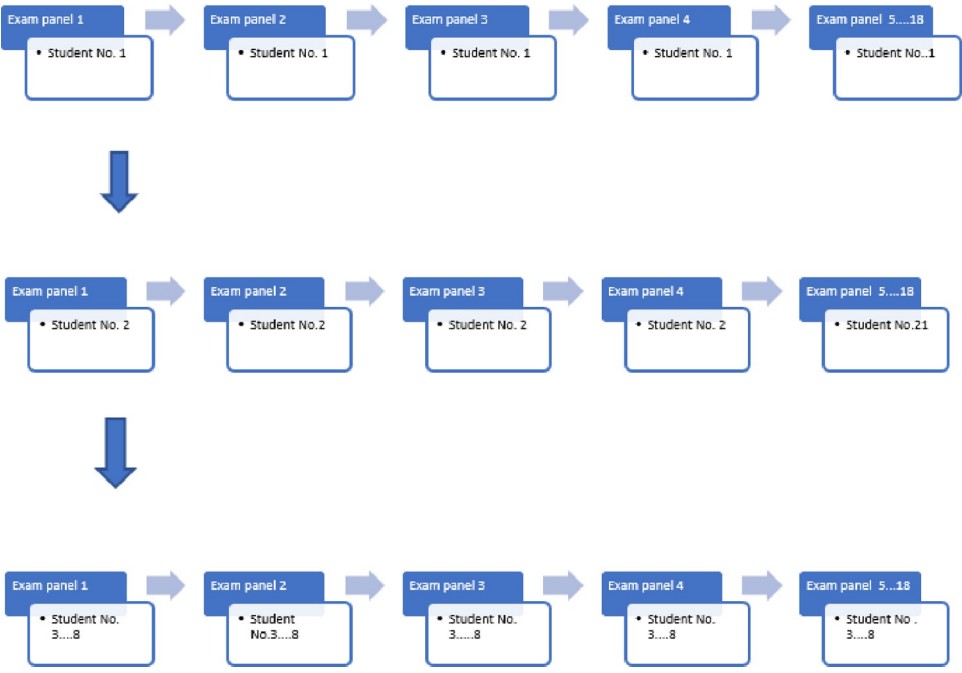

**Fig 1. Examination panels setup.** All students who assigned #1 in the 18 groups will pass through the same set of questions, the same will be applied for students #2–8. Two extra set of questions were saved to overcome any problems.

## Implementation

To overcome technical competence deficiency in some examiners, each examination panel contained one faculty member who was proficient in the Zoom application and that member was responsible for admitting the students to the Zoom meeting, controlling the waiting list and sharing the screen that show the questions to the students. Student answers were graded according to a designed checklist.

The sequence of asking questions was also the same in each examination panel starting from Surgery case followed by Medicine, Obstetrics & Gynecology, and Pediatrics.

Each exam panel examined 8 or 9 students at the same time and with the same questions for the same rounds. The question set was changed with each student to control the security of the exam (Fig 2). The time of examination for each student was 20 minutes, 5 minutes for each branch.

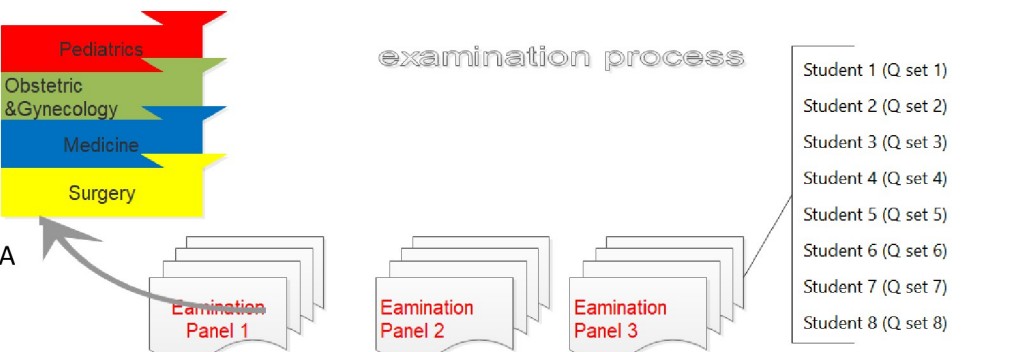

**Fig 2. Examination process.** (A) Sequence of discipline questions asked by examination panel, (B) question sets according to assigned students numbers in each examination panel i.e. 8 set of questions used by each examination panel for the 8 students.

## Stakeholder evaluation of the process

Evaluation of the online assessment process from both students and examiners perspectives was done through a questionnaire that was designed by one of the authors through literature reviews. The questionnaire was prepared and distributed to both examiners and students through a Google form. It was composed of 9 and 8 items for both examiners and students respectively that were rated on a five-point Likert scale from 1 (strongly disagree) to 5 (strongly agree). The items were related to the participants' experience and satisfaction with the setting, implementation, and environment of the online assessment. Three open-ended questions were added to the questionnaire to indicate, the most commonly liked and disliked aspects of the online examination, and suggestions to improve it. In addition to that, one more item was included in students' questionnaire about whether they faced any connection problem during the online assessment process. An overall score for each individual was calculated by summing up their Likert scale scores of all items [4]. Internal consistency reliability of the responses was evaluated by Cronbach alpha (α).

## Data analysis

The data obtained from the Google forms was shifted to another statistical software, the Statistical Package for Social Sciences (SPSS, version 25). Categorical data were summarized and presented as frequencies and percentages. Mann-Whitney and Kruskal-Wallis tests were used to compare the mean ranks of the overall satisfaction scores between categories of the students and examiners. Thematic analysis was performed for the above-mentioned open-ended questions. P-values <0.05 were regarded as significant.

## Ethical consideration

Ethical approval was obtained from the ethics committee of Hawler Medical University. (the ethical approval number: 8 on 20/4/2021) A written informed consent about providing feedback on the online assessment was obtained through Google form. Students' identity was not disclosed for ethical reasons. The confidentiality of the participants' information was maintained.

## Results

Data from 75 students and 54 examiners were available for analysis. The response rate for the electronic questionnaire was 69.4% for students and 88.5% for examiners.

The overall score for each individual was calculated, in addition to the Mean values and standard deviations. There were no significant differences in the mean ranks of the total scores between male and female students (p = 0.392), and between male and female examiners (p = 0.095), also there were no significant differences between the different specialties (p = 0.392) (Table 1).

The mean (SD) of the examiners' overall satisfaction score was 3.76 (0.67) out of 5 which was significantly (p<0.001) higher than that of students (3.1 (0.85)).

The student online assessment survey had good internal reliability (Cronbach alpha = 0.883). It was clear that the vast majority of the student responses were in the "Agree" category apart from their response to the statement of "the assessment process evaluated the clinical skills of the students" where only 10.7% agreed on it. According to 53.3% of students, the time allocated for answering questions was enough. Half of the students agreed that the examiners were professional, and the assessment questions were clear. In general, 37.3% of the students were satisfied with the assessment process (Table 2).

**Table 1. Mean satisfaction scores of the online assessment evaluation scales by gender and specialty.**

| Students | | | | Examiners | | | |
|---|---|---|---|---|---|---|---|
| | N | Mean score (SD) | P-value | | N | Mean score (SD) | P-value |
| **Gender** | | | | **Gender** | | | |
| Male | 27 | 23.9 (6.9) | 0.392* | Male | 26 | 34.46 (6.5) | 0.095* |
| Female | 48 | 25.3 (6.7) | | Female | 28 | 33.25 (5.7) | |
| | | | | **Specialties** | | | |
| | | | | Medicine | 3 | 35.7 (0.6) | |
| | | | | Obstetrics and gynecology | 20 | 34.5 (5.2) | 0.392** |
| | | | | Pediatric | 15 | 33.5 (6.1) | |
| | | | | Surgery | 16 | 33 (7.6) | |

*Mann-Whitney test

**Kruskal-Wallis test

It is worth mentioning that 33 (44%) students faced problems during the assessment process which was either electricity shutdown (21.2%), internet connection (45.5%) or both (33.3%). The majority of the students' issues were solved immediately while only 9 out of the 33 students (27.3%) waited till the end of the exam and the extra set of questions were used for them.

Table 3 shows the main themes and subthemes that emerged from the students' perception on Pros and Cons of the online assessment. The two main areas highlighted in the Pros were the question and format of the exam, and the professional behavior of the examiners. The students mostly liked the type of the questions, being practical and covering the common clinical and emergency cases reasonably (34.6%). On the other hand, the two main themes that emerged in the Cons were related to administration issues, and question and format of the assessment. The major areas that students disliked were as follows: long waiting time and delay (22.7%), unfair distribution of marks (22.7%), connection problems (20%). Other subthemes are presented in detail in Table 3.

Around one quarter (24%) of the students had no specific suggestion for improvement, 17.3% suggested another method for assessment (other than the online), 13.3% suggested better training for the students and examiners before the exam, and 13.3% of students suggested accurate timing, and to give more time. The other suggestions are mentioned in (Table 4).

Sixty-one examiners were asked to participate in the study, but only 54 (88.5%) participated. Around half (48.1%) were males. The mentioned teaching staff belonged to the

**Table 2. Students' perception about the assessment process.**

| | Disagree and Strongly Disagree | Neutral | Agree and Strongly Agree |
|---|---|---|---|
| **Indicators** | **No. (%)** | **No. (%)** | **No. (%)** |
| Students comprehensive instructions on the way of the online assessment. | 22 (29.3) | 20 (26.7) | 33 (44.0) |
| There was sufficient communication between the school administration and students. | 20 (26.7) | 26 (34.7) | 29 (38.7) |
| The format of the online assessment was acceptable. | 23 (30.7) | 25 (33.3) | 27 (36.0) |
| The examiners were professional. | 15 (20.0) | 22 (29.3) | 38 (50.7) |
| The assessment questions were clear. | 18 (24.0) | 19 (25.3) | 38 (50.7) |
| The conducted assessment evaluated students' clinical skills. | 40 (53.3) | 27 (36.0) | 8 (10.7) |
| The time allocated to answer each (medicine, surgery, OBGYN, pediatrics) was enough. | 17 (22.7) | 18 (24.0) | 40 (53.3) |
| Overall, you were satisfied with the assessment process. | 23 (30.7) | 24 (32.0) | 28 (37.3) |

**Table 3. Areas that the students like (Pros) and dislike (Cons) regarding the assessment process.**

| Themes | Subthemes | No. of responses | (%) N = 75 |
|---|---|---|---|
| **The most commonly liked aspects of the exam (Pros)** | | | |
| Questions and format of assessment | • The questions were practical and reasonable covering common clinical and emergency problems. | 26 | (34.6) |
| | • Students liked the online assessment as a new and safe solution. | 11 | (14.7) |
| Examiners behavior | • The examiners behaved professionally. | 18 | (24.0) |
| **Nothing specific.** | | 18 | (24.0) |
| **The most commonly disliked aspects of the exam* (Cons)** | | | |
| Administration issues | • Long waiting time and delay | 17 | (22.7) |
| | • Connection problem | 15 | (20.0) |
| | • Poor communication with the examiners (voice and face were not clear). | 6 | (8.0) |
| | • Unfair distribution of questions and marks as only 20% was dedicated for the final exam | 17 | (22.7) |
| | • The online assessment process | 7 | (9.3) |
| | • Preparedness and organization for the assessment process | 2 | (2.7) |
| Questions and format of the assessment | • The questions were not clear, and many of them are not emergency cases. | 9 | (12.0) |
| | • Clinical skills cannot be assessed | 5 | (6.7) |
| | • Short time allocated for the assessment process | 5 | (6.7) |
| Nothing specific | | 6 | (8.0) |

*Students may have more than one disliked area.

following departments: medicine, surgery, pediatrics, obstetrics and gynecology. The examiners' online assessment survey was found to have good internal reliability (Cronbach alpha = 0.881).

It is evident in (Table 5) that the majority of the examiners were generally satisfied with the online examination process, they agreed on all the items raised by the researchers except for 'assessment of the clinical competence' where many of them thought that the online exam could not assess the clinical competence and did not reflect real clinical practice.

Organization of the exam and teamwork were the most liked areas by the examiners (38.9% and 27.8% respectively). Four examiners (7.4%) liked the fairness of the exam, and another 7.4% liked the type of questions, being clinical and covering the curriculum. On the other hand, the most disliked areas were: the inability of the exam to reflect the clinical skills of the students" (22.2%), connection problems (18.5%), and poor organization (14.8%). Details of examiners related themes and subthemes are presented in (Table 6).

**Table 4. Students' suggestions for improvement.**

| | No. | (%) |
|---|---|---|
| No specific suggestion | 18 | (24.00) |
| To find a method of assessment, other than the online | 13 | (17.33) |
| Better training for the students and examiners before the exam | 10 | (13.33) |
| Accurate timing, and to give more time | 10 | (13.33) |
| Questions should focus on emergency conditions | 8 | (10.67) |
| Same closed-ended questions for all the students, more questions, and more examiners | 8 | (10.67) |
| Better internet and website | 5 | (6.67) |
| Others | 3 | (4.00) |
| Total | 75 | (100.00) |

**Table 5. Examiners' perception about the examination process.**

| The items | Disagree and Strongly Disagree No. (%) | Neutral No. (%) | Agree and Strongly Agree No. (%) |
|---|---|---|---|
| You were clearly informed about the assessment process | 3 (5.6) | 11 (20.4) | 40 (74.1) |
| During the assessment, the technical support by the cohost was adequate | 3 (5.6) | 12 (22.2) | 39 (72.2) |
| During the assessment, health regulation, physical distancing, and mask wearing on the day of the exam (at college) was adequate. | 6 (11.1) | 11 (20.4) | 37 (68.5) |
| Exam duration was acceptable. | 1 (1.9) | 7 (13.0) | 46 (85.2) |
| Assessment reflected real clinical practice. | 11 (20.4) | 28 (51.9) | 15 (27.8) |
| The question reflected proper sampling from the curriculum | 2 (3.7) | 18 (33.3) | 34 (63.0) |
| You were satisfied with the provided assessment checklist | 8 (14.8) | 9 (16.7) | 37 (68.5) |
| The online assessment could assess clinical competence. | 17 (31.5) | 14 (25.9) | 23 (42.6) |
| Organization of the whole process met your expectation | 3 (5.6) | 12 (22.2) | 39 (72.2) |

More than one third (36.4%) of the suggestions were about better organization, and to have more questions. Nine (16.4%) suggestions were about the training of the examiners before the exam. Eight (14.5%) suggestions were to make an on-campus exam with implementing safety measures where possible. The other suggestions are presented in (Table 7).

**Table 6. Areas that the examiners like (Pros) and dislike (Cons) regarding the assessment process.**

| | Themes | Subthemes | No. of responses | (%) |
|---|---|---|---|---|
| **The most commonly liked aspects of the exam (Pros)** | | | | |
| | Administration Issues | • Organization of the examination process | 21 | (38.9) |
| | | • Teamwork | 15 | (27.8) |
| | | • The time of the exam was known for the students | 3 | (5.6) |
| | | • A new experience in assessing the students | 1 | (1.9) |
| | Questions and format of assessment | • Fair | 4 | (7.4) |
| | | • The questions were clinical, covering the curriculum | 4 | (7.4) |
| | Safety | • Being online, so no chance to get the COVID-19 | 3 | (5.6) |
| | Nothing specific | | 3 | (5.6) |
| **The most commonly disliked aspect of the exam (Cons)** | | | | |
| | Administration Issues | • Poor organization | 8 | (14.8) |
| | | • Connection problems | 10 | (18.5) |
| | | • No social distancing | 1 | (1.9) |
| | Questions and format of assessment | • It did not reflect the clinical skills of the students | 12 | (22.2) |
| | | • Direct questions that can't differentiate between students | 3 | (5.6) |
| | | • Absence of a detailed checklist for each case | 4 | (7.4) |
| | | • The appearance of the answer key for some questions | 3 | (5.6) |
| | Examiners | • Not every examiner was involved in preparing the questions | 2 | (3.7) |
| | | • Delay of the examiners in attending on time | 1 | (1.9) |
| | Nothing specific | | 10 | (18.5) |
| | | **Total** | 54 | (100) |

**Table 7. Examiners' suggestions for improvement.**

| Suggestions | No. | (%) |
|---|---|---|
| Better organization, better preparation for the exam, and more questions. | 20 | (36.4) |
| Training of the examiners before the exam | 9 | (16.4) |
| On-campus assessment with the use of safety measures | 8 | (14.5) |
| Nothing specific | 7 | (12.7) |
| Ongoing assessment should have a role in the assessment process | 3 | (5.5) |
| Better internet connection | 2 | (3.6) |
| Involve more examiners in the process | 2 | (3.6) |
| Use of a detailed checklist for one of the departments | 1 | (1.8) |
| Use the same process of assessment in the future | 1 | (1.8) |
| Assessment by an individual examiner (not in committee) | 1 | (1.8) |
| Share the exam experience with other colleges | 1 | (1.8) |
| Total* | 55 | (100.0) |

Note. *More than one suggestion is possible for each examiner.

## Discussion

Online assessment is an unusual method of assessment for graduating medical students and proper decision and preparation with the involvement of administration and teachers is essential. The presented study represents one of the experiences of online assessment during COVID-19 lockdown.

The COVID-19 pandemic has affected medical education in many aspects. Creative ways of assessment were important to maintain the standards of medical education during the lockdown time [9], especially the assessment of clinical competencies which is a challenging area needing innovative modification [10].

As the lockdown was imposed by the government of Iraq and Kurdistan Region, to continue learning and save the academic year, the Ministry of Higher Education and Scientific Research in Iraq and Kurdistan Region decided in March 2020 to replace in-person classes with their online equivalents. These challenges were more distressing for final year medical students waiting for their graduation assessments [11].

At Hawler Medical University, College of Medicine, the college council adapted the distribution of assessments' marks, assigning 80% of the total course mark to coursework related activities, while the remaining 20% was allocated to final assessments to ensure a fair ranking of students. A similar approach was performed in Jordan [12].

As the face-to-face final OSCE was prohibited at the start of the pandemic, OSCE blueprint, instructions and structured marking schedules were adapted. Physical examination or procedural skills couldn't be assessed due to absence of simulated or real patients. Therefore, it was planned to have an objective structured viva examination instead. Questions were based on emergency oriented short case vignette from all major disciplines. Technological development and accessibility to variety of digital tools have supported the education process in COVID-19 era throughout the globe. Zoom, Google meet, Microsoft team, WebEx, . . .etc are examples. Overall, these tools enabled sharing clinical photographs, laboratory results and imaging using the screen-sharing feature, and facilitated the assessment of clinical reasoning and data interpretation domains via examiner questioning synchronously. Problem-based questions enabled the assessment of the clinical reasoning and higher-order thinking skills [13, 14]. Hawler Medical University used Zoom as a tool for the final year assessment as the staff and students had experience with it during the early COVID-19 period.

In several countries, clinical and theory examinations have been cancelled [9], postponed, or delayed or replaced by online or new methods of assessment [15, 16].

Results indicated that the overall satisfaction of both examiners and students with the conducted assessment was high with that of the examiners' being higher. Similar results were obtained by Tan et al. [17]. On the other hand, concerns about the number of questions used, the mark allocated for the final assessment, technical problems and poor nonverbal communication on video-conferencing were mainly disliked by a certain fraction of students (30.7%) and might have contributed in their non-satisfaction. According to Mak and his colleagues, it is hard to gain control on the full range of nonverbal communication through the online platform due to screen size, position of the student on the screen as well as impaired video quality due to internet problems [18]. This was also found in our students, as a good percentage of them used mobile screens for joining. The online assessment process has many challenges during implementation including connection problems [19]; variations in household internet access for both examiners and students was an expected problem especially in a country like Iraq. To minimize this concern, two points were considered, asking examiners in the examination panel to conduct the examination as a committee in the campus benefiting from the fact that the examiners (who are also physicians) had permission from the local authorities for free movement in the lockdown period and only students needed to be at home. Secondly, if any connection problem occurred for any of the students, an extra set of questions were present to be used at the end of the exam, in order not to affect the sequence of the exam as explained in detail in methodology.

Exam security was another challenge, as it was necessary to ensure that students have met the required learning outcome. Threats to assessment security can be alleviated by designing assessment methods that are resistant to challenges of cheating [20].

To overcome this problem, 18 committees conducted the exam at the same time and changed the question for each round as clarified in (Fig 1), and a Zoom room was created for every committee with enabling the waiting feature.

Less chance for cheating was declared by many students as one of the points that students liked about the process of assessment, however, it led to longer waiting time for their turn to enter the Zoom room.

To use technology in a low-technology context, adapting new approaches to solve local problems should be restricted to what is possible under the circumstances [21]. Poor technical capability of all examiners to use IT especially Zoom, as this exam was conducted at the beginning of COVID-19 (2019–2020 academic year) and lacking IT personnel in the college put a higher pressure on the assessment committee and the administrative team to find a solution. This problem was alleviated by choosing 18 examiners who were good at using technology specifically Zoom to be focal points for each examining committee.

In medical education, it is important to prepare medical students for real-life scenarios by designing authentic assessments [15, 22] to ensure the students' ability to deal with emergency case scenarios. Therefore, all case scenarios that were prepared reflected real emergency case scenarios. However, around one-third of examiners and nearly half of students disagreed that this online assessment assessed clinical skills.

The strength of this study is that it reports both examiners and students' perception on the online assessment experience. The overall satisfaction score mean of examiners were significantly higher than that of students. This agreed with Elshami et al. where satisfaction rates among students and examiners were 41.3 and 74.3% respectively [23].

Limitations: This study examines one institution experience but, it is important to share the modification of assessment process done in the pandemic time with medical educators throughout the globe especially in low resource countries. Although it was not possible to

practice a mock final online assessment by students due to uncertainty at the beginning of the pandemic, good preparation, planning and prediction of the expected problems with their solutions by focal points and effective communication with the students, had a major role in the success of the process.

In conclusion, the findings from this study suggest that assessment via alternative methods during the lockdown has enlightened examiners to the value of their use in normal situations.

There is therefore, a definite need for Information communication technology training including the effective use of distance educational tools for student-oriented teaching and assessment strategies for all faculty members. This may come as an obligatory online course in alignment with preparing new university examiners and equipping them with digital tools. An important area that needs to be addressed in future researches is to explore the perception of stakeholders on online assessments utilizing focus groups and qualitative methods.

## Supporting information

**S1 File.**
(PDF)

**S2 File.**
(PDF)

## Acknowledgments

The authors appreciate all medical students and examiners who spent their time to participate in the study.

## Author Contributions

**Conceptualization:** Nazdar Ezzaddin Alkhateeb, Ali A. Al-Dabbagh.

**Data curation:** Baderkhan Saeed Ahmed, Namir Ghanim Al-Tawil.

**Formal analysis:** Namir Ghanim Al-Tawil.

**Methodology:** Nazdar Ezzaddin Alkhateeb, Baderkhan Saeed Ahmed, Ali A. Al-Dabbagh.

**Project administration:** Nazdar Ezzaddin Alkhateeb.

**Supervision:** Ali A. Al-Dabbagh.

**Validation:** Ali A. Al-Dabbagh.

**Writing – original draft:** Nazdar Ezzaddin Alkhateeb, Baderkhan Saeed Ahmed, Namir Ghanim Al-Tawil, Ali A. Al-Dabbagh.

**Writing – review & editing:** Nazdar Ezzaddin Alkhateeb, Baderkhan Saeed Ahmed, Namir Ghanim Al-Tawil, Ali A. Al-Dabbagh.

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
