## [Decision Letter · Decision Letter 0]

27 Apr 2022

PONE-D-22-08106Students and examiners perception on virtual medical graduation exam during the COVID-19 quarantine period: a cross-sectional study.PLOS ONE

Dear Dr. Alkhateeb,

Thank you for submitting your manuscript to PLOS ONE. After careful consideration, we feel that it has merit but does not fully meet PLOS ONE’s publication criteria as it currently stands. Therefore, we invite you to submit a revised version of the manuscript that addresses the points raised during the review process.

We look forward to receiving your revised manuscript.

Kind regards,

Vijayalakshmi Kakulapati, Ph.D

Academic Editor

PLOS ONE

Journal Requirements:

2. Please provide additional details regarding participant consent. In the Methods section, please ensure that you have specified (1) whether consent was informed and (2) what type you obtained (for instance, written or verbal). If your study included minors, state whether you obtained consent from parents or guardians. If the need for consent was waived by the ethics committee, please include this information.

Reviewers' comments:

**Comments to the Author**

Reviewer #1: This article presents a description of the online assessment adopted at Hawler Medical University/Iraq during the lockdown period of the COVID-19 pandemic, mainly for final year medical students, followed by an evaluation based on questionnaires for both examiners and students. Afterwards, a discussion of the outcomes of the questionnaire is presented. 

- One cannot classify this paper as a rigorous study. The outcomes, results and conclusions are quite trivial to the extent that one could have easily predicted them. Not to add that authors themselves found that they had almost the same results with similar studies (lines 261-263 and 295-297).

- The description of the online assessment should be complemented by clear and descriptive. This not the case here. Lines 116-120 that are supposed to explain the distribution of students are not well presented, nor the figure 1. I wasn’t able to map the information presented in the paragraph to the Figure. Not to add that the figure is ‘weak’. Moreover, Figure 2 is simply not necessary; I cannot see any added value; I would remove it. Regarding Figure 3, it is a bit vague, and no explanation in the caption. “Examination” is written as “Eamination”.

- As for the statistical part, the number of the participants as mentioned in line 168, (75 students and 54 examiners) is considered not a sufficiently large one to conduct a thorough study. And more importantly, the interpretation of p-value; Are the p-values presented in Table 1 considered ‘good’ enough from a statistical perspective?

- The questionnaire is not comprehensive and questions are very general. I would expect some very specific questions related to the quality/clearness of the X-rays and other pictures/figures, diverse type of tests (blood, urine…), comprehensiveness of case studies… does the scope of the questions cover a substantial area of knowledge in each discipline?

Moreover, here is a list of some minor suggestions, authors may take into consideration:

“Conclusions and recommendations” section is very small; I would merge them with “Discussions”.

Line 22: “while” instead of “with”.

Lines 23: “…aims at sharing…” instead of “…aims to share…”. Same applies to lines 57

Line 31: usage of “had been” is to be revised.

Line 51: “have been” to be removed.

Line 55: “Our medical school”, please specify since it is the first instance, so that reader does not have to refer to the affiliations.

Line 125: “were” instead of “was” in both instances.

Line 131, a comma to be inserted after “faculties”.

Line 132, “proficient” instead of “efficient”

Usage of “had been” to be reviewed in lines 161-166.

In line 206, “Around half (48.1%) were males, and the male: female ratio was 0.92: 1.” Redundancy! either part of the sentence is to be kept not both.

Line 235, “presented” instead of “present”

Line 260, “have been” to be removed.

Line 264, “has” instead of “have”

Line 265, a semicolon instead of a comma.

In general, English should be thoroughly reviewed

Reviewer #2: In this work, the authors conducted a research work that included online questionnaire to assess virtual medical graduation exam effectiveness from both examiners’ and students’ perspectives. Results showed that majority of the examiners were generally satisfied with the online examination process compared to only around a third of the students. Both

examiners and students agreed that online examination is not suitable for assessing the physical examination skills.

I recommend that researchers behind this work:

1- Add more details on reasons why third of students were not satisfied with the online exam.

2- Use of additional methods including interviews can be helpful to collect new /augment existing data.

3- Researchers seemed to rely more on quantitative methods. I recommend to include qualitative methods as it can capture different types of data/interactions that are hard to collect with questionnaires.

4- Researchers need to add more discussion on other tools/technologies that can be used other than Zoom for examination.

---

## [Author Response · Author response to Decision Letter 0]

17 May 2022

Responses to editor and reviewers’ comments

 Comments 

Editor 1. Please ensure that your manuscript meets PLOS ONE's style requirements, including those for file naming. 

Response :Thank you for your comment. The manuscript prepared according to the journal style (figures and tables)

 2. Please provide additional details regarding participant consent. In the Methods section, please ensure that you have specified (1) whether consent was informed and (2) what type you obtained (for instance, written or verbal). If your study included minors, state whether you obtained consent from parents or guardians. If the need for consent was waived by the ethics committee, please include this information. 

Response: An additional subtitle “Ethical consideration” added to methods section and include more details on the informed written consent.

All the participants of the study were adults 

Response : The ethics statement moved to Methods section. 

Response : Many thanks for informing me. I found a retracted paper had been cited mistakenly.

Reference no. 3 changed to another reference entitled

” Huber SG, Helm C. COVID-19 and schooling: evaluation, assessment and accountability in times of crises—reacting quickly to explore key issues for policy, practice and research with the school barometer. Educ Assessment Eval Account. 2020;32: 237–270. doi:10.1007/S11092-020-09322-Y/FIGURES/7. PMID: 32837626; PubMed Central PMCID: PMC7286213”

Reviewer #1

 1. This article presents a description of the online assessment adopted at Hawler Medical University/Iraq during the lockdown period of the COVID-19 pandemic, mainly for final year medical students, followed by an evaluation based on questionnaires for both examiners and students. Afterwards, a discussion of the outcomes of the questionnaire is presented. 

Response : No comment

 2. One cannot classify this paper as a rigorous study. The outcomes, results and conclusions are quite trivial to the extent that one could have easily predicted them. Not to add that authors themselves found that they had almost the same results with similar studies (lines 261-263 and 295-297).

Response : Thank you for your comment. The study was done at the beginning of the COVID-19. All the world specially the developing countries with low resources and limited technology information were struggling to find a way for assessing final year medical students, it was necessary to share the experience of conducting an online assessment in such circumstances. 

 3. The description of the online assessment should be complemented by clear and descriptive. This not the case here. Lines 116-120 that are supposed to explain the distribution of students are not well presented, nor the figure 1. I wasn’t able to map the information presented in the paragraph to the Figure. Not to add that the figure is ‘weak’. Moreover, Figure 2 is simply not necessary; I cannot see any added value; I would remove it. Regarding Figure 3, it is a bit vague, and no explanation in the caption. “Examination” is written as “Eamination”.

Response : Thank you for your comment. We agree that the figures were not very clear, we revised the paragraph (line 116-120) to make it clearer. And added a figure legend to both figure 1 and 3, Figure 2 was removed.

The revised part is with track changes 

 4. As for the statistical part, the number of the participants as mentioned in line 168, (75 students and 54 examiners) is considered not a sufficiently large one to conduct a thorough study. And more importantly, the interpretation of p-value; Are the p-values presented in Table 1 considered ‘good’ enough from a statistical perspective? 

Response: As the reviewer mentioned the number of the participants were 75 students and 54 examiners

The sample size calculation mentioned in line 71-80, the response rate among students and examiners were 69.4% and 88.5% respectively which is considered high. 

The p value in Table 1 showed non-significant differences. 

 5. The questionnaire is not comprehensive and questions are very general. I would expect some very specific questions related to the quality/clearness of the X-rays and other pictures/figures, diverse type of tests (blood, urine…), comprehensiveness of case studies… does the scope of the questions cover a substantial area of knowledge in each discipline? 

Response: The reviewer was absolutely right regarding the questionnaire questions being very general and missed specific questions. This was the case because the aim of this study was to evaluate the first online final year assessment which was conducted in a low resource country, and to help administration for future steps and at the end to share the experience of the implemented plan with the medical education community.

 6. Conclusions and recommendations” section is very small; I would merge them with “Discussions Response: The conclusion section was merged with the discussion

 7. Line 22: “while” instead of “with”.

Lines 23: “…aims at sharing…” instead of “…aims to share…”. Same applies to lines 57

Line 31: usage of “had been” is to be revised.

Line 51: “have been” to be removed.

Line 55: “Our medical school”, please specify since it is the first instance, so that reader does not have to refer to the affiliations.

Line 125: “were” instead of “was” in both instances.

Line 131, a comma to be inserted after “faculties”.

Line 132, “proficient” instead of “efficient”

Usage of “had been” to be reviewed in lines 161-166.

In line 206, “Around half (48.1%) were males, and the male: female ratio was 0.92: 1.” Redundancy! either part of the sentence is to be kept not both.

Line 235, “presented” instead of “present”

Line 260, “have been” to be removed.

Line 264, “has” instead of “have”

Line 265, a semicolon instead of a comma.

In general, English should be thoroughly reviewed 

Response: Many thanks for your valuable corrections. All the suggested changes were done.

Reviewer#2 

1. In this work, the authors conducted a research work that included online questionnaire to assess virtual medical graduation exam effectiveness from both examiners’ and students’ perspectives. 

Response: No comment

2. Add more details on reasons why third of students were not satisfied with the online exam.

Response: Thank you for your helpful comment. Reasons are added as following paragraph

“On the other hand, concerns about the number of questions used, the marks allocated for the final assessment, technical problems and poor nonverbal communication on videoconferencing were mainly disliked by a certain fraction of students and might have contributed in their non-satisfaction. According to Mak and his colleagues it is hard to gain control on the full range of nonverbal communication through the online platform due to screen size as some students used their mobile phones, position of the student on the screen as well as impaired video quality due to internet problems[23]. This was also found in our students as a good percentage of them used mobile screens for joining the exam. ”

3. Use of additional methods including interviews can be helpful to collect new /augment existing data.

4. Researchers seemed to rely more on quantitative methods. I recommend to include qualitative methods as it can capture different types of data/interactions that are hard to collect with questionnaires. 

Response to point 3 and 4: Thank you for raising this point. we relied mainly on quantitative data but open ended questions was also used and thematic analysis was performed to extract data (Table3-6). 

5. Researchers need to add more discussion on other tools/technologies that can be used other than Zoom for examination.

Response: The requested discussion was added as the following paragraph

“Technological development and accessibility to variety of digital tools have supported the education process in COVID-19 era throughout the globe. Zoom, Google meet, Microsoft team, WebEx,…etc are examples. Overall, these tools enabled sharing clinical photographs, laboratory results and imaging using the screen-sharing feature, and facilitated the assessment of clinical reasoning and data interpretation domains via examiner questioning synchronously. Problem-based questions enabled the assessment of the clinical reasoning and higher-order thinking skills [16,17]. Hawler Medical University used Zoom as a tool for the final year assessment as the staff and students had experienced it during the early COVID-19 period.”

---

## [Decision Letter · Decision Letter 1]

23 Jun 2022

PONE-D-22-08106R1Students and examiners perception on virtual medical graduation exam during the COVID-19 quarantine period: a cross-sectional study.PLOS ONE

Dear Dr. Alkhateeb,

Thank you for submitting your manuscript to PLOS ONE. After careful consideration, we feel that it has merit but does not fully meet PLOS ONE’s publication criteria as it currently stands. Therefore, we invite you to submit a revised version of the manuscript that addresses the points raised during the review process.

ACADEMIC EDITOR:need to improve and incorporated all reviewer comments Check language corrections 

We look forward to receiving your revised manuscript.

Kind regards,

Vijayalakshmi Kakulapati, Ph.D

Academic Editor

PLOS ONE

Journal Requirements:

Additional Editor Comments (if provided):

need to check language corrections

authors are advised to address all review comments 

**Comments to the Author**

Reviewer #2: The authors addressed my comments and provided clear feedback. The qualitative analysis might need more refinement/details. The authors mentioned conducting thematic analysis but they didn't reveal patterns/themes identified in the open ended questions. How it helped and what more information it provided?

---

## [Author Response · Author response to Decision Letter 1]

15 Jul 2022

Responses to editor and reviewers’ comments

Comments 

Editor 

1. Need to check language corrections 

Response: Thank you for your comment. The manuscript underwent language editing and grammar mistakes corrected 

Response: Many thanks for informing me. 

Based on cross reference meta data link references function, some references have been removed and some were added 

The removed references were:

Ref. no. 1 

Pokhrel S, Chhetri R. A Literature Review on Impact of COVID-19 Pandemic on Teaching and Learning. High Educ Futur. 2021;8: 133–141.

Ref. no.4 

Murphy J. Assessment in medical education. Ir Med J. 2007;100: 356. Available: http://archive.imj.ie//ViewArticleDetails.aspx?ContentID=3620

Ref.no.11

Sabzwari S. Rethinking Assessment in Medical Education in the time of COVID-19. MedEdPublish. 2020;9: 1–6.

Ref. no. 12

Shehata MH, Kumar AP, Arekat MR, Alsenbesy M, Mohammed Al Ansari A, Atwa H, et al. A toolbox for conducting an online OSCE. Clin Teach. 2021;18: 236–242.

Ref.no.14

Al-Mendalawi MD. Teaching Paediatrics in Iraq Amid the COVID-19 Pandemic. Sultan Qaboos Univ Med J. 2020;20: e408

Ref. no.19

Rezaei H, Haghdoost A, Javar HA, Dehnavieh R, Aramesh S, Dehgani N, et al. The effect of coronavirus (COVID-19) pandemic on medical sciences education in Iran. J Educ Health Promot. 2021;10. doi:10.4103/JEHP.JEHP_817_20

Ref. no.24

Pettit M, Shukla S, Zhang J, Sunil Kumar KH, Khanduja V. Virtual exams: has COVID-19 provided the impetus to change assessment methods in medicine? Bone Jt open. 2021;2: 111–118.

Ref. no.26

Derbel F. Technologically-Capable Teachers in a Low-Technology Context.

The following references were added

Choi B, Jegatheeswaran L, Minocha A, Alhilani M, Nakhoul M, Mutengesa E. The impact of the COVID-19 pandemic on final year medical students in the United Kingdom: a national survey. BMC Med Educ. 2020;20: 206.

Guangul FM, Suhail AH, Khalit MI, Khidhir BA. Challenges of remote assessment in higher education in the context of COVID-19: a case study of Middle East College. Educ Assessment, Eval Account. 2020;32: 519–535.

Yapa HM, Bärnighausen T. Implementation science in resource-poor countries and communities. Implement Sci. 2018;13: 1–13.

Reviewer#2

 1. The authors addressed my comments and provided clear feedback. 

Response: No comment

 2. The qualitative analysis might need more refinement/details. The authors mentioned conducting thematic analysis but they didn't reveal patterns/themes identified in the open ended questions. How it helped and what more information it provided? 

Response: Thank you for your helpful comment. 

 Table 3 and table 6 which present thematic analysis of the open-ended questions were modified, themes and subthemes were added to it.

A paragraph was added in the results section as follows

“Table 3 shows the main themes and subthemes that emerged from the students’ perception on Pros and Cons of the online assessment. The two main areas highlighted in the Pros were the question and format of the exam, and the professional behavior of the examiners. The students mostly liked the type of the questions, being practical and covering the common clinical and emergency cases reasonably (34.6%). On the other hand, the two main themes that emerged in the Cons were related to administration issues, and question and format of the assessment. The major areas that students disliked were as follows: long waiting time and delay (22.7%), unfair distribution of marks (22.7%), connection problems (20%). Other subthemes are presented in detail in Table 3.”

---

## [Decision Letter · Decision Letter 2]

29 Jul 2022

Students and examiners perception on virtual medical graduation exam during the COVID-19 quarantine period: a cross-sectional study.

PONE-D-22-08106R2

Dear Dr. Alkhateeb,

We’re pleased to inform you that your manuscript has been judged scientifically suitable for publication and will be formally accepted for publication once it meets all outstanding technical requirements.

Kind regards,

Vijayalakshmi Kakulapati, Ph.D

Academic Editor

PLOS ONE

---

## [Editor Report · Acceptance letter]

10 Aug 2022

PONE-D-22-08106R2 

Students and examiners perception on virtual medical graduation exam during the COVID-19 quarantine period: a cross-sectional study. 

Dear Dr. Alkhateeb:

I'm pleased to inform you that your manuscript has been deemed suitable for publication in PLOS ONE. Congratulations! Your manuscript is now with our production department. 

Kind regards, 

on behalf of

Dr. Vijayalakshmi Kakulapati 

Academic Editor

PLOS ONE